**Data Availability Statement:** Data cannot be shared publicly because of the business associate agreement between the University of Texas Health

# Patterns and risk factors of opioid-suspected EMS overdose in Houston metropolitan area, 2015-2019: A Bayesian spatiotemporal analysis

Cici Bauer[1]*, Tiffany Champagne-Langabeer[2], Christine Bakos-Block[2], Kehe Zhang[1], David Persse[3], James R. Langabeer[2,4]

1 Department of Biostatistics and Data Science, School of Public Health, The University of Texas Health Science Center at Houston, Houston, Texas, United States of America, 2 ACE Research Lab, School of Biomedical Informatics, The University of Texas Health Science Center at Houston, Houston, Texas, United States of America, 3 Office of Emergency Medical Services, City of Houston Fire Department, Houston, Texas, United States of America, 4 Department of Emergency Medicine, McGovern Medical School, The University of Texas Health Science Center at Houston, Houston, Texas, United States of America

* cici.x.bauer@uth.tmc.edu

## Abstract

### Background

Opioid-related overdose deaths are the top accidental cause of death in the United States, and development of regional strategies to address this epidemic should begin with a better understanding of where and when overdoses are occurring.

### Methods and findings

In this study, we relied on emergency medical services data to investigate the geographical and temporal patterns in opioid-suspected overdose incidents in one of the largest and most ethnically diverse metropolitan areas (Houston Texas). Using a cross sectional design and Bayesian spatiotemporal models, we identified zip code areas with excessive opioid-suspected incidents, and assessed how the incidence risks were associated with zip code level socioeconomic characteristics. Our analysis suggested that opioid-suspected overdose incidents were particularly high in multiple zip codes, primarily south and central within the city. Zip codes with high percentage of renters had higher overdose relative risk (RR = 1.03; 95% CI: [1.01, 1.04]), while crowded housing and larger proportion of white citizens had lower relative risks (RR = 0.9; 95% CI: [0.84, 0.96], RR = 0.97, 95% CI: [0.95, 0.99], respectively).

### Conclusions

Our analysis illustrated the utility of Bayesian spatiotemporal models in assisting the development of targeted community strategies for local prevention and harm reduction efforts.

Science Center at Houston and the City of Houston Fire Department. The data underlying the results presented in the study are available from Office of Emergency Medical Services, City of Houston Fire Department. The contact information of the EMS office, City of Houston Fire Department is below. Additional questions of the EMS data used in this manuscript submission, including data request, can be made directly to 1801 Smith, Suite 860 Attn: Records Dept Houston, TX 77002 832.394.6860 (Office) 832.394.6882 (Fax) email: hfdemsrecs@houstontx.gov.

**Funding:** The author(s) received no specific funding for this work.

**Competing interests:** The authors have declared that no competing interests exist.

## Introduction

Although slightly decreasing from 2017 to 2018, opioid-related overdose remains a leading cause of injury-related mortality in the US, with nearly 70% of drug overdoses involving opioids [1]. According to the US Centers for Disease Control and Prevention (CDC), while deaths rates involving all opioids decreased by 2%, rates involving synthetic opioids, such as fentanyl, saw a 10% increase [1]. Furthermore, recent geospatial analysis on opioid-related overdose has shown a geographical shift from rural areas to large metropolitan areas [2]. Houston, Texas is the fourth largest and one of the most ethnically diverse counties in the US. The 2018 census projections place the total population of the city at 2.3 million [3] and the Houston metropolitan area at 6.9 million people with a growth rate of 2.02% [4]. Like most of the country, Texas has seen a significant increase in the number of overdose deaths involving synthetic opioids. However, deaths involving heroin in Texas have more than doubled and prescription opioids have remained stable [5].

Monitoring of real-time drug overdose patterns is an essential tool in combatting the epidemic; yet, federal and state surveillance systems have considerable limitations. National drug surveillance systems which rely on a top-down approach, are slow to report patterns, and can even lag behind changing overdose trends [6]. Additionally, state and local level EMS data collection systems are not standardized. EMS agencies differ drastically in resources and local EMS agencies with limited resources may be reluctant to invest in more sophisticated data collection software. Due to the variability in collection practices, researchers often rely on field notes and data captured by dispatch systems. Similarly, data from the Survey on Drug Use and Health (SAMHSA) is limited to self-report data and may underrepresent certain populations, such as low-income individuals, ethnic minorities [7], homeless, non-English speaking and technology-impaired individuals [8]. Reliance solely on emergency department (ED) reporting of opioid-related overdose events may also underestimate incidents, since overdose survivors often refuse ambulance transport [9]. Finally, the National Vital Statistics System relies on local coroner and emergency department reported causes of mortality, which may be inconsistent by locality and underreported [10].

Local population-level data, when collected and analyzed in a timely manner, can fill in the gaps left by other surveillance programs. Emergency medical services (EMS) and first responders are critical parts of the emergency care system in the US and the first phase of emergency care. There are more than 20 million EMS transports each year, and emergency 9-1-1 services offer immediate access to an operator who can provide basic life support coaching until help arrives on scene [11]. In most urban and suburban areas, EMS is a component of local fire departments (FDs) [12] and embedded into local communities where they are strategically positioned to engage with residents. Community FDs often serve as a nerve center for public health and safety education, from home safety [13] to emergency first aid [14]. Because of their close ties to the community, local EMS are the first to respond to medical emergencies and therefore the first to notice new trends, for instance, changes in opioid overdose patterns. Early recognition of these hot spots is crucial for timely and targeted intervention and investigation over time, shed light on changes in opioid use and demographics, and projections of future hot spots. The use of Bayesian spatiotemporal models, which have recently gained popularity in research on opioid-related overdose and mortality [15], could be used to provide insight into the spatiotemporal risk of opioid-related incidents and hence guide subsequent policymaking and intervention design. Model based risk estimates offer advantages over the observed risks in that the model-based approaches examine a phenomenon that exists in a particular place and point in time by observing variables that are shared by geographical locations in addition to individual characteristics. The observed EMS data often present a lot of noise,

particularly in less populated areas. Model-based approaches allow us to tease out the noise and reveal the true underlying pattern, and to identify "hot spots" of disease incidence and forecast risks. In addition, the model-based approaches allow the investigation of the potential contributing factors to the "hotspots", that can inform policy and public health intervention, and aid the resource allocation and distribution of state and federal funds to combat the opioid crisis.

## Materials and methods

### Study design

We used a retrospective cross sectional study design of prehospital data between January 1, 2015 and December 31, 2019, and mapped suspected opioid-overdose events to zip codes in the Houston metropolitan area. The study population was approximately 24% white, and 48% Hispanic. We assessed the geographical and temporal patterns in opioid-suspected overdose incidents. We developed Bayesian spatiotemporal models to identify zip code with excessive opioid-suspected incidents (i.e., hot spots), and assessed how the incidence risks were associated with zip code level socioeconomic characteristics.

### Data sources

We relied on de-identified incident location data extracted from the patient care record system of the Houston Fire Department database, which was provided under a business associates agreement between the University of Texas Health Science Center at Houston and the fire department. We defined the opioid-suspected overdose as an incident which involves administration of naloxone, an overdose reversal medication that is effective for opioids [16]. For the spatiotemporal analysis, we included a total of 2630 EMS calls on opioid-suspected overdose incidents in 84 zip codes concentrated in the densely populated inner-loop area in Houston, TX (S1 Fig) between January 1, 2015 and December 30, 2019.

### Demographic and socioeconomic variables

Zip code demographic and socioeconomic status (SES) variables were obtained from the US Census Bureau American Community Survey (ACS) 5-year estimates. ACS data were only available for years between 2015 and 2018, and so we used the data from 2018 for 2019. These variables included total population, race/ethnicity (% of White and % of Hispanic), employment (% unemployed), poverty (% living under poverty), education level (% with bachelor degree or higher), income (in dollars), insurance (% of population uninsured), occupation (% of population working as blue collar) and housing (% of renters and % of population living with crowded housing). The complete ACS variables and their summary statistics are shown in Table 1, for the 84 zip codes in Houston metropolitan areas included in this analysis. From 2015 to 2018, we observed a decrease from 5.54% to 4.12% in the average percentage of unemployment, a decrease from 19.1% to 17.3% in the average percentage of living with poverty, a decrease from 25.8% to 22.5% in the average percentage of population with no health insurance, and an increase from $29,700 to $32,700 in average per capita income. However, substantial spatial variation of these demographic and SES variables was observed at these zip codes, see Fig 1. Additional details of the SES variables are included in S1 Fig.

### Bayesian spatiotemporal models

We followed the general inseparable Bayesian spatiotemporal modeling framework by first assuming the appropriate distribution of the observed data, then assign the structures to the

**Table 1. Summary statistics of the SES variables at zip-code level for Houston metropolitan area, 2015–2018.**

| Variable | 2015 (n = 84) | 2016 (n = 84) | 2017 (n = 84) | 2018 (n = 84) |
|---|---|---|---|---|
| **Total population** | | | | |
| Mean (CV%) | 29083 (45.2) | 29425 (45.4) | 29945 (45.2) | 30237 (45.5) |
| Median [Q1, Q3] | 28154 | 28059 | 28725 | 29015 |
| | [20196, 36897] | [19957, 37888] | [20073, 38362] | [20214, 38535] |
| **White[a], %** | | | | |
| Mean (CV%) | 24.8 (94.8) | 24.5 (96.0) | 24.1 (96.4) | 23.8 (97.0) |
| Median [Q1, Q3] | 15.2 [6.48, 34.0] | 15.3 [6.20, 33.9] | 15.1 [6.10, 33.5] | 14.1 [5.80, 33.9] |
| **Hispanic, %** | | | | |
| Mean (CV%) | 47.2 (54.0) | 47.8 (54.3) | 48.2 (54.3) | 48.4 (53.9) |
| Median [Q1, Q3] | 45.3 [26.3, 69.7] | 45.8 [26.1, 70.8] | 48.2 [26.6, 69.7] | 47.8 [26.4, 69.3] |
| **Unemployed[b], %** | | | | |
| Mean (CV%) | 5.54 (44.9) | 5.01 (43.2) | 4.47 (44.2) | 4.12 (42.9) |
| Median [Q1, Q3] | 5.65 [3.80, 7.10] | 4.90 [3.30, 6.40] | 4.30 [2.90, 5.70] | 4.00 [2.90, 5.20] |
| **Bachelor[c], %** | | | | |
| Mean (CV%) | 27.2 (90.3) | 27.8 (89.3) | 28.5 (87.8) | 28.9 (87.1) |
| Median [Q1, Q3] | 16.9 [7.78, 36.5] | 16.6 [8.45, 37.2] | 17.1 [8.65, 39.7] | 17.6 [9.20, 43.1] |
| **Crowded housing[d], %** | | | | |
| Mean (CV%) | 7.28 (64.1%) | 7.07 (64.0%) | 7.07 (63.1%) | 6.97 (64.3%) |
| Median [Q1, Q3] | 6.70 [3.43, 10.6] | 6.50 [3.48, 10.0] | 6.00 [3.70, 10.3] | 5.75 [3.73, 10.0] |
| **Renters, %** | | | | |
| Mean (CV) | 50.6 (31.6) | 50.9 (31.2) | 50.7 (31.0) | 50.9 (31.5) |
| Median [Q1, Q3] | 49.5 [38.5, 60.7] | 50.5 [39.6, 60.2] | 50.2 [39.0, 59.7] | 49.3 [38.9, 60.0] |
| **Blue collar, %** | | | | |
| Mean (CV%) | 28.0 (53.3) | 28.0 (53.7) | 27.8 (53.9) | 29.2 (52.9) |
| Median [Q1, Q3] | 30.4 [15.8, 40.4] | 30.8 [15.2, 40.1] | 30.1 [15.9, 38.8] | 33.0 [15.6, 41.1] |
| **Per capita income, $** | | | | |
| Mean (CV%) | 31415 (91.0) | 32303 (92.1) | 33573 (90.6) | 34586 (89.6) |
| Median [Q1, Q3] | 18529 | 19196 | 21169 | 21088 |
| | [14971, 32853] | [15234, 33096] | [15604, 34292] | [16504, 34964] |
| **Living poverty, %** | | | | |
| Mean (CV) | 19.1 (56.4) | 18.5 (55.1) | 17.9 (56.1) | 17.3 (54.7) |
| Median [Q1, Q3] | 18.9 [10.1, 26.3] | 18.8 [10.7, 24.8] | 19.2 [10.3, 23.9] | 17.6 [10.4, 24.1] |
| **Uninsured[e], %** | | | | |
| Mean (CV) | 25.8 (42.3) | 24.6 (44.2) | 23.6 (45.7) | 22.5 (47.0) |
| Median [Q1, Q3] | 28.1 [19.3, 33.3] | 26.2 [18.5, 32.0] | 26.2 [17.1, 30.3] | 24.1 [16.2, 29.5] |

Abbreviations: SES, socioeconomic status; CV, coefficient of variation; Q1: first quartile; Q3: third quartile.

Source: Data are estimates from the 2015 to 2018 American Survey 5-year mean.

[a] Non-Hispanic Ancestry;

[b] Civilians aged 16 years and older;

[c] Persons aged 25 years or older;

[d] Defined as occupied housing units consisting of more people than rooms;

[e] Persons in the total civilian noninstitutionalized population.

spatial and temporal components. Specifically, let $Y_{it}$ denoted the observed opioid-suspect EMS overdose counts in year $t$ and zip code $i$, we assumed a Poisson distribution with relative

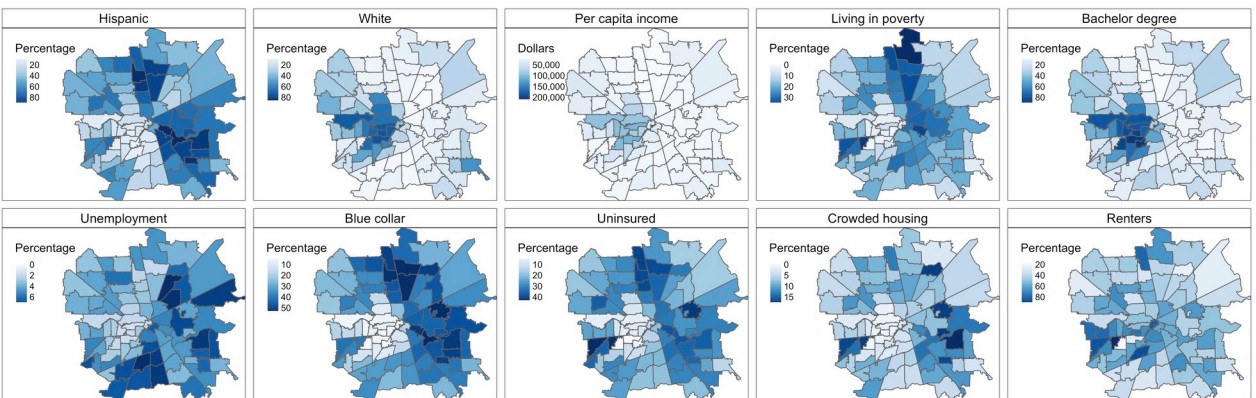

**Fig 1. Maps of the zip code social economic (SES) variables for year 2018.** SES variables were obtained from US Census American Community Survey (ACS). Geographic boundary files were downloaded from the US Census TIGER database.

risk (also known as incidence ratio) $\mu_{it}$, i.e., $Y_{it}|\mu_{it}\sim poisson(E_{it}\mu_{it})$. The offset term here was the expected count, which was defined as $E_{it} = \left(\dfrac{\sum_{it} Y_{it}}{\sum_{it} N_{it}}\right) N_{it}$ with $N_{it}$ the total population size. We chose the expected counts instead of the population size as the offset term in the Poisson model for two reasons. First, with the calculated expected counts we obtained the observed standardized incidence ratio (SIR), defined as the ratio of the observed counts to the expected counts, i.e., $SIR_{it} = Y_{it}/E_{it}$. SIRs have a straightforward interpretation, where value greater than 1 indicated more observed counts than the expected, and hence could be defined as high incidence areas or "hot spots." SIR with a value less than 1 indicated observed counts fewer than the expected, and therefore "cold spots." SIR of value 1 indicated equal observed and the expected counts. Second, we could compare observed SIR and model based relative risk (RR), as RR provided a smoothed version of SIR and was generally considered more accurate.

In the Poisson model above, we further decomposed the relative risk into the following components

$$\log \mu_{it} = \alpha + \mathbf{x}'_{it}\boldsymbol{\beta} + \varphi_i + \delta_{it}, \tag{1}$$

where $\alpha$ was the overall intercept, and $\mathbf{x}_{it}$ is the vector of zip code level covariates with coefficient vector $\boldsymbol{\beta}$. We included a random effect $\varphi_i$ to account for the spatial correlation, and a random effect $\delta_{it}$ to account for the spatiotemporal interaction. In Bayesian spatiotemporal models, there random effects were assumed to have different distributions as priors. Here we considered several popular spatial models for random effects $\varphi_i$, and compared model performance using three model selection criteria: deviance information criterion (DIC), widely applicable Bayesian information criterion (WAIC) and conditional predictive ordinate (CPO). In all three criteria, models with the smaller values of these measures were considered better [17].

The spatial models we considered here included the Besag, York, and Molliè (BYM) model [18], the Besag's proper spatial model [19] and Leroux model [20]. Details of the models can be found in their original papers but we provided a brief summary here. The BYM model considered the random effect $\varphi_i$ the summation of a spatially structured term $u_i$ and a spatially unstructured term $v_i$. The spatially structured term was assumed to have an intrinsic autoregressive (ICAR) model, where the spatial dependency between area $i$ and $j$ was encoded in the adjacency matrix $W$. The entries in $W$ were assigned value 1 if two areas shared the same geographical boundary, and 0 otherwise. The spatial unstructured term was assumed to be

independently and identically distributed. The ICAR model was also called "Besag improper" model as its joint distribution was improper (i.e., the precision matrix was singular). The Besag proper model overcame this issue by constructing a non-singular precision matrix. The third model considered here was the Leroux model where the conditional distribution of the spatial random effect was

$$\varphi_i|\varphi_{-i} = N\left\{\frac{\rho\sum_{j=1,j\neq 1}^{n} w_{ij}\varphi_j}{\rho\sum_{j=1,j\neq 1}^{n} w_{ij} + 1 - \rho}, \frac{\sigma^2}{\rho\sum_{j=1,j\neq 1}^{n} w_{ij} + 1 - \rho}\right\}. \qquad (2)$$

The parameter $\rho$ quantified the degree of spatial correlation of $\varphi_i$, with $\rho = 0$ corresponding to independence and $\rho = 1$ representing strong spatial correlation throughout the region. Residual variation not explained by spatial correlation was captured by the variance parameter $\sigma^2$. We used Integrated Nested Laplace Approximation (INLA) for Bayesian inference as an alternative to the Markov Chain Monte Carlo (MCMC), for its fast computation [21]. All analyses were performed in R using package INLA (R Studio, Boston MA).

## Results

EMS calls for opioid-suspected overdose incidence increased from 449 in 2015 to 666 in 2019 (Fig 2). We observed a slight decrease in 2017 (n = 473) compared to 2016 (n = 487), with a significant increase in calls in 2018 (n = 555) and again in 2019. In addition, we also investigated the temporal trend of incidents by month, day of the week and time of the day. Opioid-suspected overdose incidents occurred more frequent during the summer time between May and September, on Saturdays, and around the time between 7pm and 11pm.

The observed SIRs of opioid-suspected overdose EMS incidents are shown in Fig 3, where we clearly observed the spatial patterns. Darker shades indicated areas with higher risks, primarily in the south and central area of Houston. We also observed substantial changes in the spatial variation over time, which provided strong evidence to include spatiotemporal interaction in the Bayesian spatiotemporal models. Without the interaction term, we essentially assume the same spatial pattern and hotspots every year. With the interaction term in the model, we capture the changing spatial pattern over time and allow different hotspots every year. The analysis

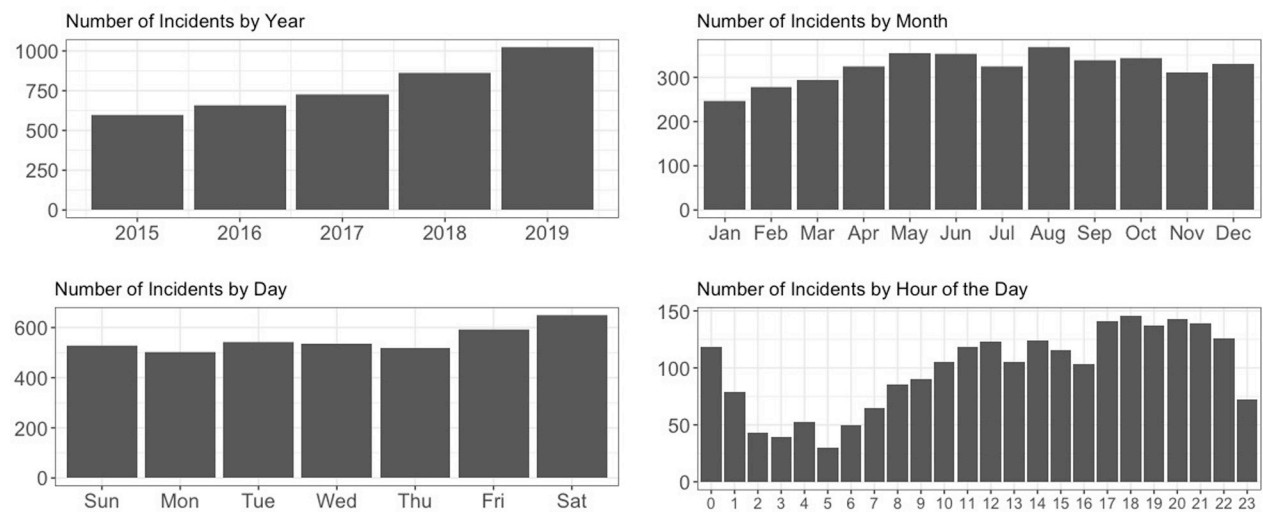

**Fig 2. Temporal characteristics of opioid-suspected EMS calls in Houston metropolitan area, 2015–2019.**

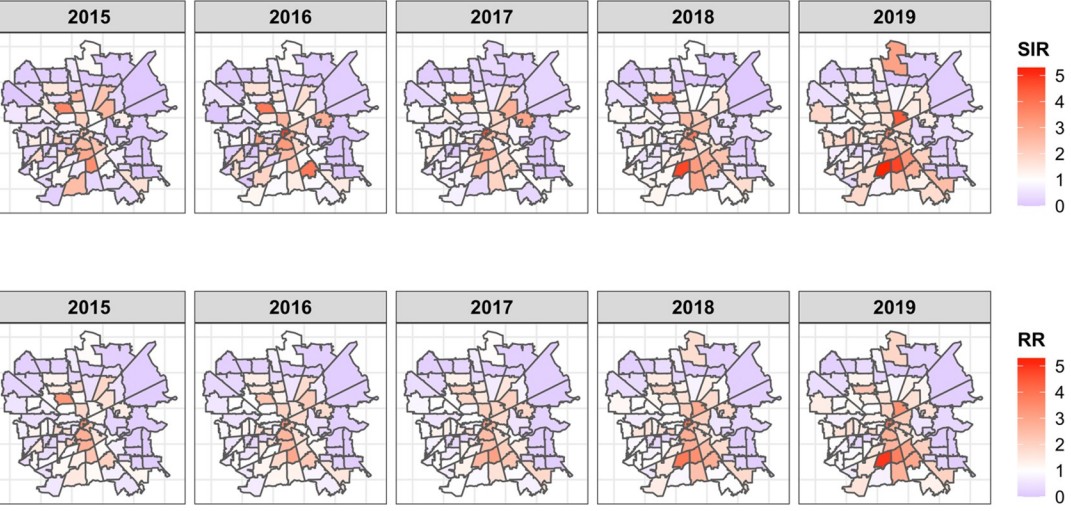

**Fig 3. Observed SIRs (top row panel) and model-based RRs (bottom row panel) of opioid-suspected EMS calls in Houston metropolitan area from 2015 to 2019.** These RRs are obtained from fitting the Bayesian spatiotemporal models using BYM spatial structure.

results from three Bayesian spatiotemporal models with different spatial structures were presented in Table 2. All model selection criteria of DIC, WAIC and CPO pointed the BYM model as the best fitting model, and therefore we made inference based on this model. Of all the SES variables investigated, we found that zip code neighborhoods with higher percentage of renters were associated with a small increase in the likelihood of higher opioid-suspected EMS relative risk (RR = 1.03; 95% CI: [1.01, 1.04]), while neighborhoods with more white population were associated with a small decease in the likelihood (RR = 0.97; 95% CI: [0.95, 0.99]). Neighborhoods with crowded housing, defined as more than one person occupying a room, appeared to have lower relative risk of opioid-suspected EMS incidents (RR = 0.9; 95% CI: [0.84, 0.96]) No other SES variables were found statistically significant. Model-based zip code level RR from the BYM model was presented in Fig 3. The spatial patterns were largely similar compared to the SIR, with elevated risks in the south part of downtown areas throughout the years, and

**Table 2. Posterior means and 95% posterior credible intervals (CIs) of the estimated coefficient of zip code level SES variables.** The regression coefficients are exponentiated to represent relative risks (RRs).

| | BYM | Besag Proper | Leroux |
|---|---|---|---|
| **% Bachelor** | 1.01 (0.98, 1.04) | 0.99 (0.96, 1.03) | 0.99 (0.96, 1.03) |
| **% Unemployed** | 0.94 (0.86, 1.02) | 0.96 (0.88, 1.04) | 0.96 (0.88, 1.04) |
| **% Uninsured** | 0.98 (0.95, 1.02) | 0.98 (0.94, 1.01) | 0.97 (0.94, 1.01) |
| **% Living under poverty** | 1 (0.97, 1.03) | 0.99 (0.96, 1.02) | 0.99 (0.96, 1.01) |
| **% Renters** | 1.03 (1.01, 1.04) | 1.02 (1.01, 1.04) | 1.02 (1.01, 1.04) |
| **% Hispanic** | 0.99 (0.98, 1.01) | 0.99 (0.98, 1.01) | 0.99 (0.98, 1.01) |
| **% White** | 0.97 (0.95, 0.99) | 0.98 (0.95, 1) | 0.98 (0.95, 1) |
| **% Crowded housing** | 0.9 (0.84, 0.96) | 0.91 (0.86, 0.97) | 0.91 (0.86, 0.97) |
| **% Blue collar** | 1.01 (0.98, 1.04) | 1.02 (0.99, 1.05) | 1.02 (0.99, 1.06) |
| **Per Capita Income (per thousand dollars)** | 1 (0.98, 1.02) | 1 (0.97, 1.02) | 1 (0.97, 1.02) |
| WAIC | 1920 | 1926 | 1926 |
| DIC | 1897 | 1900 | 1900 |
| CPO | 976 | 978 | 978 |

increasing over time. The estimated RRs were very similar across different spatial models, and a comparison of the results were included in S3 Fig.

## Discussion

Our study found that the number of EMS calls for opioid-suspected overdose significantly increased in the Houston metropolitan area between 2015 and 2019, and this trend is continuing into 2020 with 530 suspected overdose calls from January to June; a 16% increase over the same time last year. Across the nation, opioid overdose rates have seen a 54% increase in major metropolitan areas from 2016 to 2017 [1]. Originally fueled by prescription opioids, recent rises in overdoses are now driven by heroin and fentanyl, which is making the urban drug market a hotbed of overdose mortality. Applying spatiotemporal modeling to overdose incidence data, as demonstrated in this analysis can help communities struggling with overdoses, particularly those in rural areas, forecast overdose trends and develop a targeted approach to early intervention and prevention efforts. Additionally, temporal trends can be tracked and help inform staffing practices of EMS providers. Our analysis revealed more overdoses occur in the summer months, between May and September and between 7pm and 11pm. This information, combined with geospatial trends can assist municipalities with hiring and staffing practices, ensuring enough adequately trained personnel are on duty during those high-risk times.

Community strategies for understanding their own temporal and spatial patterns could be useful in positioning resources and developing neighborhood programs. The use of real-time surveillance data, from hospitals or EMS, offers specific advantages over reliance solely on death data. For example, discrepancies in reporting by state coroners was estimated in one study to that nearly 70,000 opioid-related deaths may have gone unreported since 1999 [22]. Because national statistics on drug intoxication deaths rely on data derived from state death certificates, inconsistencies in reporting from state to state can have serious implications since codes used to report causes of death are applied differently from state to state. Additionally, rural counties may lack resources for investigating drug overdose deaths and may over-use a general "presence of unspecified drugs, medicaments, and biological substances" code for reporting opioid-related overdose deaths. Similarly, overreliance on ED overdose data can similarly under-estimate overdose events, since many overdoses do not result in an ED visit. Not every 911-call results in transport to the ED [16] as survivors can refuse transport. Research has revealed a significant portion of people who inject drugs refuse EMS transport, citing fear of harassment and discrimination, potential arrest, and anticipated financial costs [23, 24]. Similarly, Good Samaritan laws have increased bystander administration of naloxone to suspected opioid overdose victims and although recommended, not every resuscitation is followed by a call to 911 [25, 26]. Although it is difficult to track, several studies have reported large percentages of overdose survivors do not get treated by EMS [27]. An estimated 25 to 50 nonfatal overdose occur for every overdose death [28] and EMS call surveillance can provide the data necessary to fill in the data gaps in identifying overdose trends.

Some communities have implemented programs that utilize EMS incident data for targeted interventions. A local Fire Department/EMS and law enforcement collaboration, in Columbus, Ohio provides free transportation to treatment facilities for overdose survivors who decline transport to the ED [29]. The REACT program also provides harm-reduction and education outreach to communities that are hard-hit by the opioid crisis [29]. In San Francisco, the DOPE program also focuses on harm reduction and conducts community outreach, using an EMS early warning system that monitors call data for any surges in overdose events [30]. Analysis of DOPE has found the program sites tend to be located in densely populated minority

communities with significant economic disparity, and who experience more overdose mortality. Furthermore, the program has demonstrated marked success in its efforts to reduce opioid-related overdose deaths [30]. A program in Houston, Texas, the Houston Emergency Opioid Response System (HEROES) uses information from EMS runs involving naloxone resuscitation to deploy a mobile response team including a peer recovery coach and licensed paramedic to engage overdose survivors into treatment [31]. Geospatial analysis of the most current data on suspected opioid-overdose calls can inform community programs on trending hot spots but can also help target specific populations that are experiencing increased overdose events by including certain demographic characteristics in the analysis. Interventions that target special populations, such as youth, females of childbearing age, and elder populations can use geospatial analysis to narrow their target areas. Spatiotemporal modeling provides advantages over monitoring real-time EMS call data. First, it helps visualize a complex and dynamic problem that fluctuates across time and space. Second, geospatial analysis can actualize changes across time spans, from days to years and can identify patterns to assist with causation research. Spatiotemporal models can help inform policy makers with budget allocation and staffing concerns. Bayesian spatiotemporal modeling approaches can examine existing trends and apply them to forecast patterns, which is useful for prevention efforts. Altogether, Bayesian spatiotemporal modeling can help EMS providers, researchers, and lawmakers refine programs and operate in a constantly changing environment.

Future policies and funding should consider incorporating geospatial modeling of first responder and other data sources within each community, to provide a more comprehensive response to the current opioid epidemic.

## Limitations

This study has several limitations. First, we focused here on one community and one major EMS provider, primarily since national EMS data at the location level do not exist. Houston, Texas is the fourth largest and most diverse city in the United States [3]; however, it may not be representative of every major metropolitan area in the US that is experiencing an influx of opioid-related overdoses, nor do our results apply to rural areas which may have different demographics and risk factors. Additionally, as with most EMS data, not all overdoses included here were necessarily opioid-related, although they were all suspected opioid overdoses indicated by the use of naloxone. EMS systems obtained data are subject to distinct limitations of convenience sampling and missing data can cause calculation errors even with imputation [32]. Finally, EMS data is not completely representative of all overdoses in a community, since patients may not always call 911 for emergency assistance.

## Conclusions

We found that several popular Bayesian spatiotemporal models were useful in identifying the socioeconomic covariates that were potentially associated with the high EMS incidence risk. We noted clear patterns emerging geographically, as well as over time. Areas with high rates of renters had higher overdose incidence risk, while crowded housing and larger proportion of white citizens had lower rates. Bayesian models could be useful to develop targeted community strategies for local prevention and harm reduction efforts.

## Supporting information

**S1 Fig. Map of the study region which includes 84 zip code areas within Hwy 610 belt in Houston, Texas.** Counts are the total opioid-suspected overdose from EMS calls between 2015

and 2019.
(DOCX)

**S2 Fig. Correlation of the zip code SES variables within study region.**
(DOCX)

**S3 Fig. Comparison of SIR and RRs estimated from different spatial models.**
(DOCX)

## Author Contributions

**Conceptualization:** Cici Bauer, Tiffany Champagne-Langabeer, James R. Langabeer.

**Data curation:** Cici Bauer, David Persse, James R. Langabeer.

**Formal analysis:** Cici Bauer, Kehe Zhang.

**Investigation:** Cici Bauer, Tiffany Champagne-Langabeer, Christine Bakos-Block, James R. Langabeer.

**Methodology:** Cici Bauer.

**Visualization:** Cici Bauer, Kehe Zhang.

**Writing – original draft:** Cici Bauer, James R. Langabeer.

**Writing – review & editing:** Cici Bauer, Tiffany Champagne-Langabeer, Christine Bakos-Block, Kehe Zhang, David Persse, James R. Langabeer.

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
