## [Decision Letter · Decision Letter 0]

15 Oct 2020

PONE-D-20-23109

Patterns and Risk Factors of Opioid-suspected EMS Overdose in Houston Metropolitan Area, 2015-2019: a Bayesian Spatiotemporal Analysis

PLOS ONE

Dear Dr. Bauer,

Thank you for submitting your manuscript to PLOS ONE. After careful consideration, we feel that it has merit but does not fully meet PLOS ONE’s publication criteria as it currently stands. Therefore, we invite you to submit a revised version of the manuscript that addresses the points raised during the review process.

I concur with both reviewers who indicate that while the analyses are very well done, there is a need for further scientific justification for the analyses conducted.  The manuscript could also benefit from additional consideration to how the spatiotemporal models can inform overdose response and future interventions (as one reviewer alludes to). 

Please submit your revised manuscript by November 13, 2020. If you will need more time than this to complete your revisions, please reply to this message or contact the journal office at plosone@plos.org. Please include the following items when submitting your revised manuscript:

We look forward to receiving your revised manuscript.

Kind regards,

Nickolas D. Zaller

Academic Editor

PLOS ONE

Journal Requirements:

2.We note that you have indicated that data from this study are available upon request. PLOS only allows data to be available upon request if there are legal or ethical restrictions on sharing data publicly. For information on unacceptable data access restrictions, please see http://journals.plos.org/plosone/s/data-availability#loc-unacceptable-data-access-restrictions.

3.PLOS requires an ORCID iD for the corresponding author in Editorial Manager on papers submitted after December 6th, 2016. Please ensure that you have an ORCID iD and that it is validated in Editorial Manager. To do this, go to ‘Update my Information’ (in the upper left-hand corner of the main menu), and click on the Fetch/Validate link next to the ORCID field. This will take you to the ORCID site and allow you to create a new iD or authenticate a pre-existing iD in Editorial Manager. Please see the following video for instructions on linking an ORCID iD to your Editorial Manager account: https://www.youtube.com/watch?v=_xcclfuvtxQ

4. Please ensure that you refer to Figure 3 in your text as, if accepted, production will need this reference to link the reader to the figure.

5.We note that [Figure(s) 1 and 3] in your submission contain [map/satellite] images which may be copyrighted. All PLOS content is published under the Creative Commons Attribution License (CC BY 4.0), which means that the manuscript, images, and Supporting Information files will be freely available online, and any third party is permitted to access, download, copy, distribute, and use these materials in any way, even commercially, with proper attribution. For these reasons, we cannot publish previously copyrighted maps or satellite images created using proprietary data, such as Google software (Google Maps, Street View, and Earth). For more information, see our copyright guidelines: http://journals.plos.org/plosone/s/licenses-and-copyright.

1.    You may seek permission from the original copyright holder of Figure(s) [1 and 3] to publish the content specifically under the CC BY 4.0 license. 

<h1>** **</h1>

Reviewers' comments:

Reviewer's Responses to Questions

**Comments to the Author**

1. Is the manuscript technically sound, and do the data support the conclusions?

Reviewer #1: Yes

Reviewer #2: Yes

2. Has the statistical analysis been performed appropriately and rigorously? 

Reviewer #1: Yes

Reviewer #2: Yes

3. Have the authors made all data underlying the findings in their manuscript fully available?

Reviewer #1: Yes

Reviewer #2: No

4. Is the manuscript presented in an intelligible fashion and written in standard English?

Reviewer #1: Yes

Reviewer #2: Yes

5. Review Comments to the Author

Reviewer #1: This study used Bayesian spatiotemporal models to identify zip code-level characteristics that are associated with high rates of opioid overdoses in a large metropolitan area. Below, I outline a number of areas where this manuscript might be improved. Overall, I think the paper could be improved by making it clearer how the spatiotemporal models could help with local efforts to combat the opioid epidemic over more real-time displays of the actual EMS data.

Abstract

-Methods: It would be helpful to include some general demographic information about the cohort along with the explicit statement that this is a cross-sectional analysis.

-Conclusions: The conclusion of the abstract could be improved by including a clinically relevant insight from the results section. Overall, what do the results tell us about how to focus our community overdose strategies?

Introduction

-A sentence about the availability of real-time EMS data could be helpful in knowing how easy it would be for localities to adopt the use of that data. Is most EMS data across the country lagging in time?

Methods

-Cross-sectional study design?

-This sentence isn’t clear to me: “Data from 2015 to 2018 were included; however, data from 2019 was not published and so we used data from 2018 for 2019.”

-It might help someone like me who is not fluent in Bayesian models to include a citation on why these three model selection criteria were used.

RESULTS:

-I think there is an extra ‘9’ in this sentence: “We observed a slight decrease in 2017 9 (n=473) compared to 2016 (n=487), 123 with a significant increase in calls in 2018 (n=555) and again in 2019.”

-Are there other zipcode level data sources other than ACS that could be incorporated into the analysis (e.g., distressed community index)?

Discussion

-Do we have any citations on how often an overdose occurs where EMS is not called?

-In the discussion of the programs that use EMS data (e.g., DOPE), it would be good to expound upon the paragraph talking about why geospatial modeling could provide a more comprehensive response to the epidemic. How could this be more helpful? What specific aspects?

-I would include more of a discussion of the months and time of day that overdoses occur. This could help with EMS staff allocation.

-It is still unclear to me how this model would be more beneficial than real-time EMS data? What does this offer over systems to display real-time data on overdoses?

-Expound upon this sentence from the Results in the Discussion: “We also observed substantial changes in the spatial variation over time, which provided strong evidence to include spatiotemporal

interaction in the Bayesian spatiotemporal models.” How is this beneficial?

Limitations

-I would consider adding a limitation that this analysis does not likely represent rural areas.

-Do you have any estimates of how much the convenience sample issue may cause errors?

-Do you have an estimate of how often overdoses occur but are not called into 911? If so, that would be good to add with the last limitation.

Reviewer #2: This spatial ecological panel analysis uses Emergency Medical Service data for opioid overdoses in Houston, aggregated within 84 ZIP codes over 5 years (n = 420 ZIP code-years). The authors use Bayesian conditional autoregressive models to assess the geographic distribution of opioid overdoses in relation to demographic characteristics, as measured using American Community Survey data. The statistical analyses are very well constructed and presented, and I commend the authors for their meticulous work and clear explanations. Nevertheless, I think the basic framing of the paper needs some attention, and there are some other issues that detract from an otherwise elegant statistical analysis.

Major issues:

1. The main problem is that aim of the paper (i.e. to demonstrate the use of Bayesian spatial methods as applied to this particular outcome) is not a terribly compelling scientific justification for the work. These models have been around for some time, and they’ve been applied to many different outcomes. I don’t think there would be any doubt that they can be used for opioid overdoses, of that there will be geographic variation in overdoses. Absent a convincing rationale, the analysis becomes a simple description of the spatial distribution of the outcome, and an atheoretical examination of associations with demographic predictors. I suggest the following steps:

a. Propose and test a clear theoretical mechanism that would cause opioid overdose to concentrate in areas with specific social conditions.

b. Justify theoretically the spatial and temporal scale of the analyses (i.e. why ZIP codes? Why years?)

c. Consider adding other areal characteristics that might be theoretically related to the outcome, beyond basic demographics.

2. The demographic characteristics are likely to be correlated with one another. I suggest presenting a correlation matrix, or using some data reduction approach to combine these variables in a meaningful way.

3. The emphasis in the Introduction section on EMS data as a rapid and comprehensive data source seems to set up a surveillance study. A few lines to justify the data source is plenty for an analytic study that assessed associations between an exposure and an outcome.

Minor issues:

o Line 53 = which “university”?

o The methods section should note early that the temporal partition is by year – this seems to be missing.

o Line 62: “Data from 2015 to 2018 were included; however, data from 2019 was not published and so we used data from 2018 for 2019.” I’m not sure what that sentence means.

o It’s important to note whether the ACS data are 1-year estimates or 5-year estimates.

o Line 89: Why is the smoothed rate considered more “accurate”

o Equation 1: Is there a reason you didn’t include a time trend? Or perhaps there’s a reason I’m not following that you wanted that variation to be captured in the error term?

o Line 118: Should say that INLA approximates the MCMC

o Figure 3: Y-axis should be on the log scale for ratios

6. PLOS authors have the option to publish the peer review history of their article (what does this mean?). If published, this will include your full peer review and any attached files.

Reviewer #1: No

Reviewer #2: No

---

## [Author Response · Author response to Decision Letter 0]

29 Dec 2020

A response to reviewers letter that address each point raised by the academic editor and reviewer(s) has been submitted as a separate file with name 'Response to Reviewers.docx'.

---

## [Decision Letter · Decision Letter 1]

1 Feb 2021

Patterns and Risk Factors of Opioid-suspected EMS Overdose in Houston Metropolitan Area, 2015-2019: a Bayesian Spatiotemporal Analysis

PONE-D-20-23109R1

Dear Dr. Bauer,

We’re pleased to inform you that your manuscript has been judged scientifically suitable for publication and will be formally accepted for publication once it meets all outstanding technical requirements.

Kind regards,

Nickolas D. Zaller

Academic Editor

PLOS ONE

Additional Editor Comments (optional):

Reviewers' comments:

Reviewer's Responses to Questions

**Comments to the Author**

1. If the authors have adequately addressed your comments raised in a previous round of review and you feel that this manuscript is now acceptable for publication, you may indicate that here to bypass the “Comments to the Author” section, enter your conflict of interest statement in the “Confidential to Editor” section, and submit your "Accept" recommendation.

Reviewer #1: All comments have been addressed

2. Is the manuscript technically sound, and do the data support the conclusions?

Reviewer #1: Yes

3. Has the statistical analysis been performed appropriately and rigorously? 

Reviewer #1: Yes

4. Have the authors made all data underlying the findings in their manuscript fully available?

Reviewer #1: Yes

5. Is the manuscript presented in an intelligible fashion and written in standard English?

Reviewer #1: Yes

6. Review Comments to the Author

Reviewer #1: There is one typo I noticed in the sentence in the Discussion below (overdoes):

"An estimated 25 to 50 nonfatal overdoes occur for every overdose death (23) and EMS call surveillance can provide the data necessary to fill in the data gaps in identifying overdose trends."

7. PLOS authors have the option to publish the peer review history of their article (what does this mean?). If published, this will include your full peer review and any attached files.

Reviewer #1: No

---

## [Editor Report · Acceptance letter]

11 Feb 2021

PONE-D-20-23109R1 

Patterns and Risk Factors of Opioid-suspected EMS Overdose in Houston Metropolitan Area, 2015-2019: a Bayesian Spatiotemporal Analysis 

Dear Dr. Bauer:

I'm pleased to inform you that your manuscript has been deemed suitable for publication in PLOS ONE. Congratulations! Your manuscript is now with our production department. 

Kind regards, 

on behalf of

Dr. Nickolas D. Zaller 

Academic Editor

PLOS ONE